# Zipoy-Voorhees Gravitational Object as a Source of High-Energy Relativistic Particles

**Bobur Turimov** [1,2,3,4,*] and **Bobomurat Ahmedov** [1,5,6]

1   Ulugh Beg Astronomical Institute, Astronomy St. 33, Tashkent 100052, Uzbekistan; ahmedov@astrin.uz
2   College of Engineering, Akfa University, Kichik Halqa Yuli St. 17, Tashkent 100095, Uzbekistan
3   Research Centre for Theoretical Physics and Astrophysics, Institute of Physics, Silesian University in Opava, Bezručovo nám. 13, CZ-74601 Opava, Czech Republic
4   Webster University in Tashkent, Alisher Navoiy 13, Tashkent 100011, Uzbekistan
5   Physics Faculty, National University of Uzbekistan, Tashkent 100174, Uzbekistan
6   Tashkent Institute of Irrigation and Agricultural Mechanization Engineers, Kori Niyoziy, 39, Tashkent 100000, Uzbekistan
*   Correspondence: bturimov@astrin.uz

**Abstract:** The Zipoy-Voorhees solution is known as the $\gamma$-metric and/or $q$-metric being static and axisymmetric vacuum solution of Einstein field equations which becomes strong curvature naked singularity. The metric is characterized by two parameters, namely, the mass $M$ and the dimensionless deformation parameter $\gamma$. It is shown that the velocity of test particle orbiting around the central $\gamma$-object can reach the speed of light, consequently, the total energy of the particle will be very high for a specific value the deformation parameter of the spacetime. It is also shown that causality problem arises in the interior region of the physical singularity for the specific value of the deformation parameter when test particles can move with superluminal velocity being greater than the speed of light that might be an additional tool for explaining the existence of tachyons for $\gamma > 1/2$ which are invisible for an observer.

**Keywords:** $\gamma$-spacetime; naked singularity; high-energy relativistic particle; orbital velocity



## 1. Introduction

The mechanisms of the production of high-energy cosmic rays of energy $10^{20}$ eV are quite an interesting and important task in modern astrophysics. There are several mechanism to explain high-energy cosmic rays, for instance, Penrose process (PP) [1], Blandford-Znajek (BZ) mechanism [2] and Banados-Silk-West (BSW) effect [3]. It is also widely believed that sources of such high-energy cosmic rays are generated by the acceleration of the charged particle in the presence of the external magnetic field in the vicinity of the gravitational compact objects, such as Quasars and Blazars, etc... (See, e.g., [4–8]). A very high centre energy production from the collisions of two particles in the vicinity of the regular black holes has been discussed in Refs. [9,10].

In the present research note, we have demonstrated that high-energy neutral particles can be generated due to the deformation of the spacetime around the gravitational compact objects which can be in the form of the naked singularity. The naked singularity is a simple example for the deformed spacetime which is a hypothetical astrophysical compact object containing physical singularity and uncovered with an event horizon. The Penrose cosmic censorship claims that gravitational singularity may not be observable. However, the existence of naked singularity is important from observability of gravitational collapse of objects to infinity density. One simple example of the naked singularity is given by Zipoy-Voorhees spacetime, so-called the $\gamma$-metric [11,12] which is also known as the $q$-metric [13–17].

In Ref. [18], the contribution of the scalar field in the $\gamma$-spacetime has been investigated. Note that these spacetime metrics belong to Weyl's class of solutions [19–21]. Several

studies in the $\gamma$-metric can be found in the recent literature: optical appearance [22], the accretion process [10,23], shadows [24,25], geodesics motion [26], spinning particle motion [27] and charged particles dynamics [28,29].

In the present paper, we derive the expression for the orbital velocity by considering particle motion around the gravitational object described by the Zipoy-Voorhees spacetime [11,12].

The paper is organized as follows. In Section 2, we provide a very detailed derivation of the orbital velocity and energy of test particle in an arbitrary spacetime. In Section 3, we determine the velocity and energy of particle orbiting around compact objects described by the $\gamma$-metric. Finally, in Section 4, we summarize obtained results.

Throughout the paper, we choose the $(-, +, +, +)$ signature for the metric tensor and spherical coordinates $x^{\alpha} = (t, r, \theta, \phi)$. Greek (Latin) indices run from 0 to 3 (1 to 3).

## 2. Formulation

The Lagrangian for test particle of mass $m$ is given by

$$\mathscr{L} = \frac{1}{2} m g_{\alpha\beta} \dot{x}^{\alpha} \dot{x}^{\beta} , \qquad \dot{x}^{\alpha} = \frac{dx^{\alpha}}{d\lambda} , \tag{1}$$

where $g_{\alpha\beta}$ is the metric tensor, $\lambda$ is an affine parameter, $\dot{x}^{\alpha}$ is the four-velocity normalized as, $g_{\alpha\beta} \dot{x}^{\alpha} \dot{x}^{\beta} = -1$, that allows to write the following expression:

$$g_{tt} \dot{t}^2 (1 - v^2/c^2) = -1 , \tag{2}$$

where $v$ is the relative three-velocity of particle measured in a frame of a local observer defined as, $v^2 = v_{\hat{r}}^2 + v_{\hat{\theta}}^2 + v_{\hat{\phi}}^2$, and the orthonormal components are given as [30–32]

$$v_{\hat{r}} = \frac{dr}{dt} \sqrt{\frac{g_{rr}}{-g_{tt}}} , \qquad v_{\hat{\theta}} = \frac{d\theta}{dt} \sqrt{\frac{g_{\theta\theta}}{-g_{tt}}} , \qquad v_{\hat{\phi}} = \frac{d\phi}{dt} \sqrt{\frac{g_{\phi\phi}}{-g_{tt}}} . \tag{3}$$

The conserved quantities, namely, the energy and angular momentum of test particle are given by

$$P_t = -E = mc^2 g_{tt} \dot{t} , \qquad P_{\phi} = L = m g_{\phi\phi} \dot{\phi} . \tag{4}$$

Hereafter eliminating, $\dot{t}$, from Equations (4) and (2), the classical expression for the energy as well as angular momentum of relativistic particle in the framework of general relativity (GR) can be expressed as [30–32]

$$E = \frac{mc^2 \sqrt{-g_{tt}}}{\sqrt{1 - v^2/c^2}} , \qquad L = \frac{m v_{\phi} \sqrt{g_{\phi\phi}}}{\sqrt{1 - v^2/c^2}} . \tag{5}$$

On the other hand, from the standard calculations one can easily obtain the effective potential of test particle in the $\gamma$-spacetime. Using the normalization of the four-velocity of particle, i.e., $g_{\alpha\beta} \dot{x}^{\alpha} \dot{x}^{\beta} = -1$, and taking into account the expressions in (4), one can obtain the expression for the effective potential in the form (See, e.g., [18,33]):

$$V_{\text{eff}} \equiv -g_{tt} \left( 1 + \frac{(L/mc)^2}{g_{\phi\phi}} \right) = \left( \frac{E}{mc^2} \right)^2 + g_{tt} \left( g_{rr} \dot{r}^2 + g_{\theta\theta} \dot{\theta}^2 \right) . \tag{6}$$

Considering the motion of particle at the equatorial plane $\theta = \pi/2$ and $\dot{\theta} = 0$ and by applying the conditions $\dot{r} = \ddot{r} = 0$, the critical values for the conserved quantities can be found as

$$E = \frac{mc^2 \sqrt{-g_{tt}}}{\sqrt{1 - \frac{g'_{tt}}{g'_{\phi\phi}} \frac{g_{\phi\phi}}{g_{tt}}}} , \qquad L = \frac{mc \sqrt{g_{\phi\phi}}}{\sqrt{1 - \frac{g'_{tt}}{g'_{\phi\phi}} \frac{g_{\phi\phi}}{g_{tt}}}} \sqrt{\frac{g'_{tt}}{g'_{\phi\phi}} \frac{g_{\phi\phi}}{g_{tt}}} , \tag{7}$$

where prime denotes the derivative with respect to the radial coordinate.

Now, comparing the energy expressions (5) and (7), one can express the orbital velocity of particle in the form:

$$v = v_\phi = c \sqrt{\frac{g'_{tt}}{g'_{\phi\phi}} \frac{g_{\phi\phi}}{g_{tt}}} = c \sqrt{\frac{\partial_r \ln g_{tt}}{\partial_r \ln g_{\phi\phi}}} , \tag{8}$$

which can be easily determined for any spacetime geometry. It is easy to show that the orbital velocity of particle at the photonsphere always equals to speed of the light, i.e., $v = c$. Because the position of the photonsphere can be found from the fact that the denominator of the energy expression (7) should be zero.

## 3. Results and Discussions

So far we have shown the detailed derivation of the expressions for the orbital velocity and total energy of particle. Now one can apply these results for a given spacetime. As a result, we choose the well-known $\gamma$-metric which is represented as [28,29]

$$g_{tt} = -\left(1 - \frac{2M_*}{\gamma r}\right)^\gamma , \qquad g_{\phi\phi} = r^2 \sin^2\theta \left(1 - \frac{2M_*}{\gamma r}\right)^{1-\gamma} , \tag{9}$$

where $\gamma$ is the deformation parameter of the spacetime geometry, $M_* = GM/c^2$, $M$ is the mass of the central object, $G$ is the gravitational constant. In the case when $\gamma = 1$, the metric (9) reduces to the Schwarzschild one, while for the large values of the $\gamma$ parameter, one can have

$$\lim_{\gamma \to \infty} g_{tt} = -e^{-\frac{2M_*}{r}} , \qquad \lim_{\gamma \to \infty} g_{\phi\phi} = r^2 \sin^2\theta e^{\frac{2M_*}{r}} . \tag{10}$$

Note that we are not interested in other two components of the metric tensor because they are not involved in the previous analyses.

Hereafter performing simple algebraic manipulations, the orbital velocity of test particle (8) in the $\gamma$-metric can be found as

$$v = c \sqrt{\frac{\gamma M_*}{\gamma r - M_* - \gamma M_*}} . \tag{11}$$

It is interesting to determine orbital velocity in the characteristic radii, namely, the innermost stable circular orbit (ISCO) $r_\pm$, photon-sphere $r_{ph}$ and singularity $r_s$ can be found as [28,29]

$$r_\pm = M_* \left(3 + \frac{1}{\gamma} \pm \sqrt{5 - \frac{1}{\gamma^2}}\right) , \tag{12}$$

$$r_{ph} = M_* \left(2 + \frac{1}{\gamma}\right) , \qquad r_s = \frac{2M_*}{\gamma} , \tag{13}$$

which leads to statement that the orbital velocity of test particle at the ISCO position is determined as

$$v_\pm = c \sqrt{\frac{\gamma}{2\gamma \pm \sqrt{5\gamma^2 - 1}}} . \tag{14}$$

Consequently, the energy expression for test particle (7) at the stable circular orbit takes a form:

$$\frac{E_{\pm}}{mc^2} = \sqrt{\frac{2\gamma \pm \sqrt{5\gamma^2 - 1}}{\gamma \pm \sqrt{5\gamma^2 - 1}} \left(\frac{3\gamma - 1 \pm \sqrt{5\gamma^2 - 1}}{3\gamma + 1 \pm \sqrt{5\gamma^2 - 1}}\right)^{\gamma}}. \tag{15}$$

Our analyses show that there is no problem with the velocity of particle at the positive ISCO, $v_+$, for any value of the $\gamma$ parameter. However, at negative ISCO, it even might be greater than the speed of light for the specific values of the $\gamma$ parameter. Figure 1 shows the dependence of the orbital velocity of the particle, $v_{\pm}$, at the ISCO positions from the $\gamma$ parameter. One can see that the maximum value for the velocity at the positive ISCO is $v_+ \leq c/\sqrt{2} \simeq 0.7c$ for $\gamma = 1/\sqrt{5}$, while for the large value of the $\gamma$ parameter, it will be $v_+ = c\sqrt{\sqrt{5} - 2} \simeq 0.486c$. Interestingly, as we see from Figure 1, the velocity of particle at the negative ISCO may reach the speed of light, i.e., $v_- = c$ at $\gamma = 1/2$, while it is greater than the speed of light for $\gamma > 1/2$ that leads to causality problem. Here, one might conclude that there is a tool/mechanism for the possible generation of tachyon which is a purely hypothetical particle with the superluminal velocity being greater than the speed of light. However, one has to keep in mind that tachyon might exist only beyond the physical singularity i.e. inside the singularity, and is consequently invisible for an observer.

A similar situation can happen with the energy of particle $E_{\pm}$. Figure 2 illustrates the dependence of the particle's energy at the ISCO position from the $\gamma$ parameter. The analyses show that the positive solution of the energy $E_+$ of particle is always finite for any value of the $\gamma$ parameter. For $\gamma = 1$, the energy of particle at the positive ISCO will be $E = (2\sqrt{2}/3)mc^2$, however, at the negative ISCO, i.e., $E_-$, it tends to infinity for $\gamma = 1/2$ or

$$\lim_{\gamma \to 1/2} E_- = \infty, \tag{16}$$

while in this case, the particle's velocity tends to the speed of light. If one slightly increases the $\gamma$ parameter from the value of $\gamma = 1/2$ then the energy of particle $E_-$ will be imaginary with $v_- > c$ which produces a causality problem (corresponds to tachyon). To compare the results we also present the energy as well as velocity of particle in the Schwarzschild and extreme Reissner-Nordström (RN) spacetimes. The obtained results are listed in Table 1.

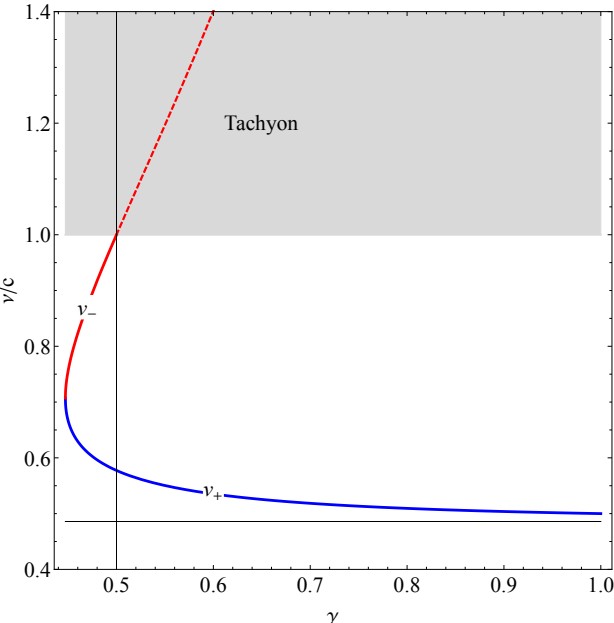

**Figure 1.** The orbital velocity of particle at the closest stable orbit as functions of the $\gamma$ parameter.

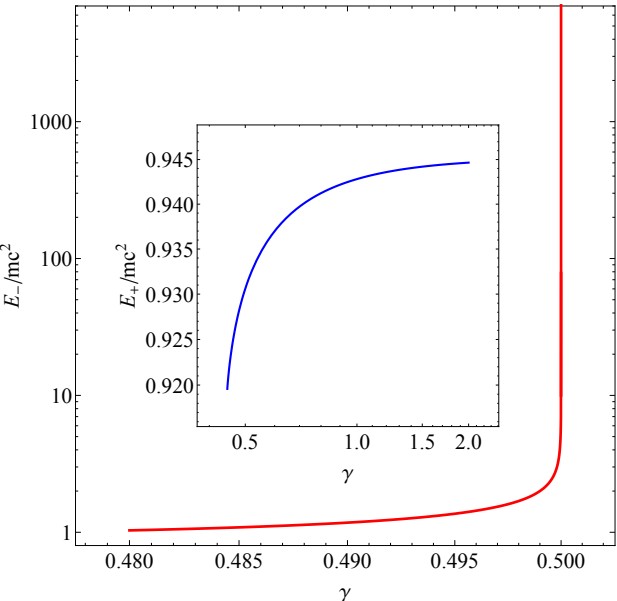

**Figure 2.** The total energy of particle at the ISCO position as functions of the $\gamma$ parameter.

**Table 1.** The orbital velocity and energy of test particles orbiting at the ISCO position around the central objects described by the Schwarzschild, extreme Reissner-Nordström RN and $\gamma$-metric.

| Spacetime | $-g_{tt}$ | $g_{\phi\phi}/r^2 \sin^2\theta$ | $v/c$ | $E/mc^2$ |
|---|---|---|---|---|
| Schwarzschild | $1 - 2M_*/r$ | $1$ | $1/2$ | $\sqrt{8}/3$ |
| Extreme RN | $(1 - M_*/r)^2$ | $1$ | $\leq 1/\sqrt{3}$ | $\leq \sqrt{27/32}$ |
| $\gamma$-metric | $(1 - 2M_*/\gamma r)^\gamma$ | $(1 - 2M_*/\gamma r)^{1-\gamma}$ | $\sim 1$ | $\sim \infty$ |

As we mentioned before the dependence of the characteristic radii from the $\gamma$ parameter has been discussed by the number of authors [27–29]. However, there are still open questions on the behavior of the characteristic radii, in particular, when the value of the parameter is around $\gamma = 1/2$. Here, one can clearly answer this question. The asymptotic behaviours of characteristic radii in (12) and (13) reads

$$\lim_{\gamma\to\infty} r_\pm = \left(3 \pm \sqrt{5}\right)M_*\,, \qquad \lim_{\gamma\to\infty} r_{\rm ph} = 2M_*\,, \qquad \lim_{\gamma\to\infty} r_* = 0\,. \tag{17}$$

Figure 3 illustrates the dependence of the characteristic radii from the $\gamma$ parameter. As one can see from the figure that the positive ISCO position is well defined for any value of $\gamma$ parameter, while the negative ISCO disappears in the range $(1/2, 1)$ of the $\gamma$ parameter and lies in the inner region of the singularity, and for $\gamma > 1$, it again appears with the lower values than the position of the photonsphere. As we have already found that the orbital velocity at the photon-sphere is equal to the speed of light. From Figure 3, one can see that $\gamma = 1/2$ is the critical point for all radii and below this value, the photon-sphere disappears while above this value the ISCO position $r_-$ disappears. Test particle with a velocity greater than the speed of light might exist at the inner region of the singularity. Finally, one can conclude that the ISCO position $r_-$ is responsible for a relativistic particle for $\gamma < 1/2$ and for tachyon at $\gamma > 1/2$ which is invisible for the observer.

It is worth noting that the structure of the $\gamma$-metric is very similar to the well-known Janis-Newman-Winnicour (JNW) spacetime, in particular, at an equatorial plane they are identical. However, in the case of the JNW metric, the $\gamma$ parameter arises due to the massless-scalar field. The performed calculations will be the same in this metric. That is why we also easily can follow our approach and apply it to JNW spacetime.

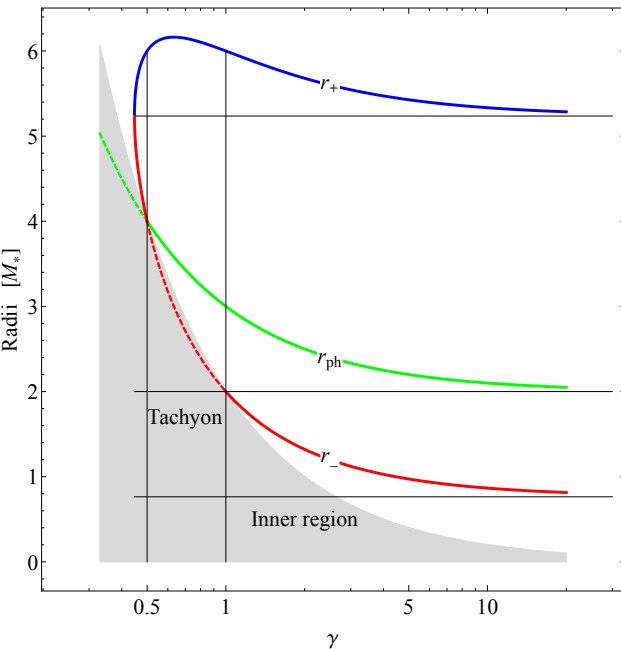

**Figure 3.** Dependence of the characteristic radii, namely, ISCO, photonsphere and singularity of the spacetime from the $\gamma$ parameter.

## 4. Summary

In the present research note, we have derived the explicit expression for the orbital velocity of test particle in arbitrary spacetime. The classical expression for the energy of relativistic particle has been derived in the frame of general relativity.

It is shown that the orbital velocity of massless particle (photon) is equal to the speed of light in arbitrary spacetime. We underline that test particle orbits at the ISCO position around the gravitational object with relativistic velocity, in particular, with half of the speed of light, i.e., $v = c/2$ around the Schwarzschild black hole, while with $v \leq c/\sqrt{3}$ around the maximally charged black hole (the extreme RN spacetime).

It is investigated that particle may orbit around the gravitational compact object described by the $\gamma$-metric with the velocity of $v \leq c/\sqrt{2}$ at the positive ISCO position shown in Equation (12), even it can orbit with a velocity greater than the speed of light at the negative ISCO position $r_-$ that might be an additional tool for explaining the existence of tachyon interior region of the singularity.

It is also shown that the energy of particle in the spacetime of $\gamma$-metric is finite at the positive ISCO position, while it is extremely high at the negative ISCO position (i.e., $E_- \to \infty$) when $\gamma = 1/2$. That is why one can conclude that the gravitational object described by $\gamma$-metric might be the source of the high-energy relativistic particles.

**Author Contributions:** Conceptualization, B.T. and B.A.; methodology, B.T.; software, B.T.; validation, B.A.; formal analysis, B.A.; investigation, B.T.; writing—original draft preparation, B.T.; writing—review and editing, B.T. and B.A.; visualization, B.T.; supervision, B.A. Both authors have read and agreed to the published version of the manuscript.

**Funding:** This research was funded by the Abdus Salam International Centre for Theoretical Physics under the Grant No. OEA-NT-01.

**Institutional Review Board Statement:** Not applicable.

**Informed Consent Statement:** Not applicable.

**Data Availability Statement:** No data available in this article.

**Acknowledgments:** This work was supported by the internal grant SGS/12/2019 of SU, Czech Republic and by Grants of Uzbekistan Ministry for Innovative Development.

**Conflicts of Interest:** The authors declare no conflict of interest.

## Abbreviations

The following abbreviations are used in this manuscript:

GR　　General Relativity
ISCO　Innermost stable circular orbit
RN　　Reissner-Nordström
JNW　Janis-Newman-Winnicour

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
