# Peer review of "Zipoy-Voorhees Gravitational Object as a Source of High-Energy Relativistic Particles"

_galaxies, doi:10.3390/galaxies9030059_

Round 1

Reviewer 1 Report

In this paper, Zipoy-Voorhees gravitational objects are investigated as a source of high-energy relativistic particles. The Zipoy-Voorhees solution is the static and axisymmetric vacuum solution of Einstein field equations which becomes strong curvature naked singularity. The discussions could be interesting and the mathematical results might be helpful for the related future works. Thus, if the following points are reconsidered carefully, this paper could be worthy of being published.

1. There exist past related works on astrophysics around naked singularity of black holes in the literature. By comparing with these preceding studies, the new ingredients and significant progresses of this work should be stated more explicitly and in more detail. That is, the differences between this paper and the past ones should be described in more detail and more clearly. This is the most crucial point in this review. 

2. It is stated that the velocity of test particle orbiting around central g-object can reach to the speed of light, consequently, the total energy of the particle will be very high for the specific value of the deformation parameter of the spacetime. Since the test particle will be massive, it seems this result must be due to a kind of apparent effect of special relativity, otherwise the conformal invariance must be broken. This point should be mentioned. 

3. Moreover, it is argued that causality problem arises in interior region of the physical singularity if test particles can move with superluminal velocity, and therefore the existence of tachyons in interior region of the singularity could be suggested. Related to the 2nd point, what are the physical reasons why test particle can have such a superluminal velocity leading to the breaking the causality? 

4. Finally, it is recommended that the wordings and grammar of English should be rechecked throughout the present manuscript. 

Reviewer 2 Report

 In this paper,  the authors used the Zipoy-Voorhees solution to study the velocity of test particle orbiting around central object can reach to the speed of light, consequently, the total energy of the particle will be very
high for the specific value of the deformation parameter of the spacetime. Moreover, they show that the causality problem arises in interior region of the physical singularity for the specific value of the deformation parameter when test particles can move with superluminal velocity being greater
than the speed of light that might be additional tool for explaining the existence of tachyons in  interior region of the singularity which are invisible for an observer. What makes significant the manuscript is the discussion on the gravitational object which described by gamma-metric might be the source of the high-energy relativistic particles.

Reviewer 3 Report

In this manuscript named “Zipoy-Voorhees gravitational object as a source of high-energy relativistic particles” the authors compute the velocity and energy of test particles orbiting around central sources described by the Zipoy-Voorhees solution. They show that this velocity can reach or even exceed the speed of light for specific values of the deformation \gamma-parameter. Causality problem arising in the interior region of the physical singularity is then analyzed in relation to the possible existence of tachyons.

I do have some reservations about the manuscript, which I would like to see addressed before a possible publication. Please, find my queries below:

1) The authors develop their analysis within the single-particle picture. For such strong high-energy phenomena, I would rather expect a field theoretical treatment by considering in Eq.(1) the Lagrangian for a field. To what extent are the present results affected in that case? The authors should comment on this point.

2) English is quite fine, but some effort is still needed to improve it.

3) In the abstract, the authors refer on several occasions to “the specific value of the deformation parameter”.  At this level, however, such value is not yet known to the reader, so I suggest the use of the indefinite article “a specific value”.

4) The effective potential in Eq.(4) has never been define before. For the sake of clarity, I suggest to provide some more details before introducing this equation.

5) In the last limit of Eq.(17) the authors perhaps mean r_s rather than r_*

6) The authors should better explain this point: looking at the red plot in fig.1, it seems that we can have tachyons for any \gamma > 1/2. However, looking at the negative ISCO plot in fig.3, it seems that the only region allowed for tachyons is 1/2 < \gamma <  1. What happens for \gamma >1? Can we still talk of tachyons or not? I would say no, otherwise we would see tachyons in the outer region of the singularity (since for \gamma > 1, r_- > r_s). Please, comment on this.

I would be able to reconsider the manuscript after the above issues have been properly addressed.

Round 2

Reviewer 1 Report

The authors' answers to the review report are appreciated very much. 
In the revised manuscript, the points suggested in the review report 
have been reconsidered. Thus, this paper can be accepted for publication 
in Galaxies

Reviewer 3 Report

The authors have quite adequately satisfied my queries. The manuscript is now suitable for publication.